# Emerging Role of Adiponectin/AdipoRs Signaling in Choroidal Neovascularization, Age-Related Macular Degeneration, and Diabetic Retinopathy

**DOI:** 10.3390/biom13060982

**Published:** 2023-06-13

**Authors:** Mayank Choubey, Puran Bora

**Affiliations:** 1Department of Foundations of Medicine, New York University Long Island School of Medicine, Mineola, NY 11501, USA; 2Pat & Willard Walker Eye Research Center, Department of Ophthalmology, Jones Eye Institute, University of Arkansas for Medical Sciences, 4301 West Markham, Little Rock, AR 72205, USA

**Keywords:** adiponectin (APN), adiponectin receptors (AdipoRs), retinal pigment epithelium, age-related macular degeneration (AMD), choroidal neovascularization (CNV)

## Abstract

Age-related macular degeneration (AMD), a leading cause of irreversible blindness in adults, may result in poor central vision, making it difficult to see, read, and drive. AMD is generally classified in either dry or wet types. Milder cases of dry AMD may progress to geographic atrophy (GA), leading to significant visual disability; wet, or neovascular AMD, which involves choroidal neovascularization (CNV), can lead to complete loss of central vision. Adiponectin (APN) discovery in the mid-1990’s and, subsequently, its two cognate receptors (AdipoRs) in the early 2000s have led to a remarkable progress in better understanding metabolic disorders, as well as metabolism-associated ocular pathology. APN/AdipoRs signaling plays a central role in a variety of molecular and cellular physiological events, including glucose and lipid metabolism, whole-body energy regulation, immune and inflammation responses, insulin sensitivity and retinal cell biological functions. This review is an amalgamation of recent information related to APN/AdipoRs in the pathophysiology of retinal diseases and furthers its association with AMD and diabetic retinopathy. Additionally, we present our original research, where we designed control peptide and CNV inhibitory peptide from the globular region of APN to see the effect of these peptides on the mouse model of laser-induced CNV. The inhibitory peptide (APN1) inhibited CNV by more than 75% while the control peptide did not inhibit CNV.

## 1. Introduction

Adiponectin (APN), a ~30 kDa regulatory obesity-associated peptide hormone, encoded by the ADIPOQ gene, is copiously secreted and mostly derived from white adipose tissue (WAT). APN regulates several key physiological functions in the human body, playing a vital role in glucose and fatty acid metabolism [1,2]. Additionally, it functions to control glucose and lipid homeostasis [3], whole-body energy regulation [4], immune and inflammatory responses [5], as well as aging and metabolic alterations [6,7,8]. Interestingly, APN provides a protective function to ocular tissue [8,9]. The retina is one of the most metabolically active tissues in the human body, at times surpassing the metabolic rate of the brain; this can lead to blood vessel growth and regression in reaction to these high metabolic demands. APN has also shown a protective function in several retinal illnesses, including diabetic retinopathy (DR) [9], choroidal neovascularization (CNV) secondary to age-related macular degeneration (AMD) [10], and other retinal pathologies [11].

AMD is generally classified as wet type and dry type. Wet age-related macular degeneration (AMD) is a prominent cause of irreversible blindness or complete loss of central vision in older adults. Choroidal neovascularization (CNV) is the hallmark of wet type AMD. Dry AMD does not typically cause complete blindness, but it may result in poor central vision, which can make it harder to see, read, and drive. When advanced, dry AMD progresses to geographic atrophy (GA) or even to wet AMD, both of which lead to severe vision loss.

Retinal hypoxia, which triggers increased metabolic demand, leads to signaling pathways that attempt to recruit new vascular supplies, resulting in the neovascularization of the eye [10]. The literature suggests an alteration in the most abundant circulating adipokines, APN and leptin, which are physiologically implicated in the metabolic modulation of various tissues and may contribute to the progression of neovascular eye diseases [9]. Additional evidence suggests that the increased secretion of the adipocyte-derived hormone leptin is associated with poor energy homeostasis, the increased oxidative stress of vascular endothelial cells (ECs), and subsequent endothelial cell dysfunction, leading to retinopathy [12]. Another key metabolic modulator, derived chiefly from adipocytes ‘APN,’ also contributes to various metabolic disorders in the retina. Concentrations of circulating APN are linked with diabetic retinopathy [13,14], growth and progression of premature retinopathy [15], and age-related macular degeneration [16]. As manifested by studies on laser-induced choroidal neovascularization [17,18] and the oxygen-induced proliferative retinopathy rodent model, there is suppression of pathological vascular proliferation upon increasing the levels of circulatory APN [19]. By focusing on prevalent investigation and relevant research, we will analyze this emerging field of APN/AdipoRs and their role in retinal neovascular disorders.

## 2. Milestones in APN/AdipoRs Research: History of APN and Elucidation of Its Physiological Functions

### 2.1. APN/AdipoRs: Historical Overview, Structural, and Circulational Forms

APN was discovered simultaneously by four different laboratories in the mid-1990’s and separately named 30 kilo Dalton (KDa) adipocyte complement-related protein (Acrp30), AdipoQ, Adipose Most abundant gene transcript 1 (apM1), and 28 kDa Gelatin Binding Protein (GBP28) [20,21,22,23]. APN receptors, also known as AdipoRs, on the other hand, were not discovered until decades later [24]. This breakthrough discovery by Yamauchi and colleagues in 2003 demonstrated a molecular mechanism of adiponectin activity after they isolated two closely related membrane-bound receptors, i.e., AdipoR1 and AdipoR2, from human skeletal muscle, utilizing expression cloning vectors [25]. 

The APN gene is located on chromosome 3q27, spanning a 16 kilo basepair (kbp). The human APN gene encodes a 244 amino acid (aa) secretary peptide (247 aa for mouse ortholog) of size ~30 KDa. The primary structure of APN consists of the Carboxy (C)-terminal globular domain and Amino (N)-terminal collagen-like domain [26]. Both human and mouse AdipoR1 comprise 375 aa proteins and have a predicted molecular mass of 42.4 kDa, while human AdipoR2 protein comprise 299 aa (and mouse, 311 aa), and has a predicted molecular mass of 35.4 kDa. Additionally, both human and mouse AdipoR1 and AdipoR2 share 96.8% and 95.2% of their identity, respectively. Interestingly, despite these AdipoRs being transcribed from two distinct genes, they share 67% of their sequence identity. Both the receptor isoforms exist to be integral membrane proteins receptors, while N-terminus is internal-facing and the C-terminus is external-facing, which is contrary to the topology and function of all other reported G protein-coupled receptors (GPCRs) [27]. Studies have shown the variation of these receptors in binding affinity for various forms of APN. AdipoR1 shows high affinity for a globular APN, while AdipoR2 shows an intermediate affinity for both globular and full-length APN. APN along with AdipoRs are ubiquitously expressed in different type of tissues [28]. In addition to AdipoR1 and AdipoR2, APN also works by binding to receptor T-cadherin, although, at this time, its role appears less critical when compared to AdipoRs [29].

The obesity hormone, APN, forms a complex structure and circulates in the plasma as a low molecular weight (LMW) trimer, a middle molecular weight (MMW) hexamer, and a high molecular weight (HMW) oligomer (Figure 1). These various forms of APN differ in their biological activity. HMW APN was shown to be the biologically active form of the obesity hormone [30]. Under certain circumstances, the HMW form has been shown to be a better insulin sensitizer compared to other LMW forms. In addition to its complex structure, APN is glycosylated, a post-translational modification that is essential to sustain its activity. Its circulation level ranges from 3 µg/mL to 30 µg/mL in humans, as well as in rodents, which is one of the most abundant adipokine in the plasma [31,32].

### 2.2. APN/AdipoRs: Tissue Distribution, Signaling, Physiological Actions, and Pathophysiological Significance

Most predominantly secreted and produced by adipocytes, APN is expressed on various potential sites and exerts beneficial effects on several metabolically active organs and cells [33,34], including liver parenchymal cells, i.e., hepatocytes [35], skeletal muscle and myocytes [36], brain [37], vasculature [38], male and female reproductive organs [6,39], and ocular tissue [9,40]. Both AdipoR1 and AdipoR2 are widely and abundantly expressed, not only in the skeletal muscle and liver tissues, but also in the macrophages [41], hypothalamus [42], WAT [43], reproductive tissues [44,45], and the retina [46]. It has been identified that AdipoRs function as a crucial mediator of APN signaling in both in vitro and in vivo investigations [8]. 

APN interacts with its two well-known specific cell-surface receptor isoforms (AdipoR1 and AdipoR2) [24]. Moreover, AdipoR1 is more firmly involved in the activation of the AMP-activated protein kinase (AMPK) pathway and further regulates the inhibition of hepatic glucose production together with increased fatty-acid oxidation, whereas AdipoR2 is involved in the activation of the peroxisome proliferator-activated receptor alpha (PPARα) nuclear receptor pathways, which triggers fatty-acid oxidation and inhibits tissue inflammation and oxidative stress [47]. The expression of AdipoRs in the insulin-targeted metabolic organs, such as the liver and skeletal muscle, significantly increases during fasting conditions, compared to refed conditions in the rodent model. Furthermore, in vitro studies have revealed that the expression of AdipoRs is reduced by insulin via the phosphoinositide 3-kinase/FoxO1-dependent pathway [48].

The circulating concentration of APN and the expression of AdipoR1 and AdipoR2 in the metabolically active organs decline in obese and diabetic individuals but increase in healthy and lean individuals [49,50,51]. One study showed sex-related differences in APN synthesis, with males having a lower level than females. The levels of APN are significantly reduced in obese patients with Type II DM and infertility [52]. Generally, insulin stimulates, whereas tumor necrosis factor alpha (TNF-α) inhibits the expression and release of APN [53]. In addition, APN is known for its involvement in the control of metabolism and insulin sensitization and it also modulates inflammatory responses by reducing the production and activity of TNF-α and interleukin-6 (IL-6) in the macrophages by suppressing NF-κB activation [54]. APN is also involved in the regulation of various bodily functions, such as glucose utilization, lipid synthesis, energy homeostasis, vasodilation, and retinal function [15,55]. APN deficiency leads to insulin resistance, glucose intolerance and hyperlipidemia in mice [56,57]. APN along with other obesity-derived adipokines play major roles in the interaction between metabolism and reproduction, and may be implicated in the detrimental effect of aging on male reproductive functions [7,39]. Supplementation with APN may represent an important therapeutic strategy for obesity-linked reproductive disorders, including infertility [6,58].

The in vitro knockdown experiments with APN in skeletal muscles illustrate the pivotal role of AdipoR1 in mediating several processes, such as β-oxidation, PPAR activation, glucose uptake, and AMPK activation [59]. The activity of AMPK in the liver is increased by AdipoR1, hence influencing the process of gluconeogenesis. Synchronously, AdipoR2 is responsible for scavenging reactive oxygen species (ROS), the activation of the nuclear receptor PPAR along with β-oxidation, mediated by its downstream target genes. A few intracellular facets of AdipoR1 signaling have been illustrated by Mao and his colleagues, utilizing yeast 2-hybrid approach [60]. They illustrated the interaction of the intracellular AdipoR1 fragment with the Adaptor protein, phosphotyrosine, interacting with the PH domain and leucine zipper 1 (APPL1), bearing a phosphotyrosine-binding domain, as well as an adaptor molecule with a plecktrin homology domain. The noteworthy point here is that this very interaction is phosphotyrosine-dependent but through tyrosine-phosphorylation-independent pathways. The ubiquitously expressed APPL1 justifies the widespread pertinence of adiponectin signaling in various metabolic tissues. Apart from AdipoRs, APPL1 may also interact with other receptors. As such, in vitro investigations exhibit that the overexpression of APPL1 results in enhanced glucose uptake, β-oxidation APN-stimulated AMPK activation. Additionally, APPL1 may play a role in the exchange between insulin signal transduction and APN/AdipoRs signaling cascade [60]. Hence, it can be propounded that these two hormones can have synergistic activity through the alike adaptor molecule. 

The development of nonalcoholic fatty liver disease (NAFLD) and steatohepatitis (NASH) with fibrosis and inflammation in obese fa/fa Zucker rats that were fed a high-cholesterol/ high-fat diet manifests in the role of AdipoRs in the modulation of the metabolism of fatty acids in the liver. There was a significant decrease in the expression of both isoforms, AdipoR1 and AdipoR2, during the NASH condition, which in turn is associated with decreased PPARα and AMPKα 1/2. Specific tissues also determine these outcomes as liver AdipoR1 participates in activating AMPK, whereas AdipoR2 is actively involved in PPARα activation, triggering increased insulin sensitivity [61].

Due to its ability to establish communication with other organs and lipid retention capacity, APN plays an indispensable role in the maintenance of lipid and glucose homeostasis. In one study, the overexpression of adiponectin/AdipoRs showed numerous beneficial effects, such as reduction in visceral fat and amelioration of liver inflammation and fibrosis [62].

## 3. APN/AdipoRs as a Regulator of Metabolism-Associated Retinal Disease

The studies regarding the presence of APN in the retina [46] and eyes [63] focus particularly on its role in DR, retinopathy of prematurity, photoreceptor integrity, retinitis pigmentosa, hypoxia-mediated retinal neovascularization, and AMD. As mentioned previously, the retina is one of the most demanding metabolic tissues of the body, with photoreceptors containing more mitochondria than any other cell [64]. The retina receives the nutritional/oxygen supply through the blood vessels and the premature forfeiture of blood vessels triggers hypoxia and energy substrate deficiency, which is considered a major factor in initiating angiogenesis in retinal tissue. Hypoxia decreases the activity of propyl hydroxylase, which is known to expeditiously reduce the hypoxia-inducible factor (HIF)-1 protein under optimal oxygen tension. Increased levels of HIF-1 protein trigger the expression of angiogenic factors, such as vascular endothelial growth factor A (VEGF-A). Certain metabolically driven HIF-1-independent pathways can also modulate the expression of VEGF-A [65]. VEGF-A assists the proliferation of blood vessels in the pursuit of reinstating the oxygen and energy substrate supply to the retina. However, these newly developed blood vessels are altered, and having abnormal morphology, may actually cause damage to the sensitive retinal tissue [66], advancing to blindness in severe cases. 

Neurodegenerative eye diseases cause symptoms such as fuzzy, blurry, or distorted vision, and the activation of the APN/AdipoRs signaling pathway has shown to be a potent neuroprotective mechanism, aiding in this condition and improving vision. Both APN and AdipoRs are present in various retinal cells. Upon binding to AdipoRs, and subsequently, activating the downstream molecular pathway, APN exerts its action, and its expression can be detected in the retina. APN is secreted predominantly by adipose tissues [43], but it may also be produced locally by the retina [46] and brain [37]. It enters the circulation and passes the blood–brain barrier with ease.

Many pathophysiological conditions, such as high glucose, dyslipidemia and mitochondrial irregularities, may disturb retinal functions and may contribute to retinal vascular disorders [67]. A prime activator of glycolysis, 6-Phosphofructo-2-kinase/fructo-2, 6-bisphosphatase isoform 3, plays a crucial function in angiogenesis [68]. The alteration of the glucose metabolic enzymes in the polyol pathway act as a protective role for the retina against retinal dysfunction and neovascularization [69,70]. Moreover, another way to inhibit vessel sprouting is to block the rate-limiting enzyme in the fatty acid oxidation process, camitine palmitoyl transferase 1 [71].

## 4. Current Insight into the Pathophysiological Role of APN in DR and AMD

A few studies have described the protective role of APN/AdipoRs pathways. One such study was performed with Type 2 DM patients in Japan. Using a laser Doppler velocimetry technique, it was determined that the retinal blood vessel diameter, as well as retinal blood velocity and flow in males positively correlated with plasma APN levels but not in females. A better lifestyle balance or use of drugs that lead to increased plasma APN levels may have revealed a new path to develop novel therapeutic approaches in the treatment of diabetes [72]. High glucose has been identified as the most important predisposing factor for angiogenesis in DR. Results from a previous study demonstrated the involvement of APN in dysregulated autophagy and retinal angiogenesis. In addition, it was shown that APN had a protective effect on the high glucose-induced damage of RF/6A cells, and further, it prevented the high glucose-induced angiogenesis of RF/6A cells by inhibiting the autophagy pathway [73].

This cumulative research suggests that APN may be a novel and highly effective therapeutic target to treat DR-related angiogenesis. Specifically, it may effectively reduce retinal neovascularization as it inhibits basal tube formation in the primary human cell culture of retinal microvascular ECs, umbilical vein macrovascular ECs, and choroidal ECs. This effect is also linked with the obstructed VEGF role that contributes to angiogenesis in DR [74,75]. The closest paralog of APN, a cytokine C1q/TNF-related protein-9, can preserve the blood retinal barrier (BRB), which minimizes the inflammatory response seen in DR in diabetic db/db mice [40].

In addition to DR, APN-related therapeutics may be effective in treating the vision-threatening sequelae of AMD. AMD, which includes two common phenotypes, i.e., Dry AMD and Wet AMD, is characterized by drusen formation. This material, consisting of protein, fat, minerals, and other debris, is tightly bound to proteins to form ball-like structures. Over time, the drusen damage the retina, causing irreversible changes to the cells. Dry AMD can progress and change to Wet AMD, which is characterized by the formation of new vessels from the choroid; these vessels invade the retinal pigment epithelium (RPE) and the subretinal space, causing central blindness (Figure 2, A Schematic Diagram). Mallordo and coworkers have hypothesized the differential functions of APN in ocular diseases, showing inhibitory effects in the proliferation and migration of RPE cells [76]. Moreover, Osada and coworkers demonstrated the impact of AdipoR1 deletion in the abnormal lipid metabolism in the retina, as well as retinal neurodegeneration, specifically utilizing AdipoR1 KO mice. Furthermore, they utilized in situ hybridization and revealed that AdipoR1 mRNA was strongly expressed in the photoreceptor inner segment (PIS) and weak staining in the inner retinal layers in 4-week-old control mouse retinas. Retina AdipoR1 expression appears to be vital for an elongase of very long chain fatty acids’ (ELOVL2) induction, which may be an essential step to supply sufficient docosahexaenoic acid (DHA) for appropriate photoreceptor cells’ functioning and survival [77].

Bushra and coworkers observed the inhibitory role of APN in the ECs adhesion and extracellular matrix organization, with a parallel improvement in the barrier function, thereby ameliorating the high glucose-induced damages on human microvascular retinal endothelial cells (HMRECs) [78]. Another study utilizing streptozotocin (STZ)-induced diabetic mouse model investigated the effect of APN on the advancement of early retinal vascular damage. The immunofluorescence data showed T-cadherin-dependent localization of APN in the vascular endothelium of retinal arterioles, which was progressively diminished during diabetes. Such decrease in retinal APN expression accompanied the early features of DR, represented by the increased permeability of vessels, and was prevented by glucose-lowering therapy with dapagliflozin, a selective sodium–glucose co-transporter 2 inhibitor [79]. Additionally, APN deficiency resulted in severe vascular permeability under relatively short-term hyperglycemia, together with a significant increase in the vascular cellular adhesion molecule-1 (VCAM-1) and a reduction in claudin-5 in the retinal endothelium [79].

Neovascular AMD is the primary cause of irreversible legal blindness in the elderly. This progressive eye disease affects the macula, the central yellow-pigmented area of the retina, containing color-sensitive rods that are vital for sharp, central vision. Evidence from the literature suggests the crucial role of APN in ameliorating neovascularization in AMD [17,18]. Out of the three major categories of accepted CNV animal models (i.e., laser-induced, surgically induced, and genetically engineered animal models), the laser-induced rodent model of CNV has more advantages compared to others because it is comparatively simple to create, inexpensive, more reproducible, and highly efficient. This model may be utilized in conjunction with the immune manipulation model to simulate various inflammatory responses associated with AMD. 

Mice lacking the very low-density lipoprotein receptor knockout (Vldlr KO) exhibited pathological retinal angiomatous proliferation, a condition that also affects people with AMD. Omega 3-long-chain-polyunsaturated-fatty-acid (ω3-LCPUFA), a supplementary ingredient added in the food in the form of docosahexaenoic acid (DHA) and eicosapentaenoic acid (EPA), is known to suppresses laser-induced CNV in controlled mice, whereas this suppression is abolished in the APN KO mice. Moreover, in the retinas of Vldlr KO mice, the ω3-LCPUFA can also increase AdipoR1 expression and inhibit neovascularization. The clinical and experimental evidence suggests that ω3-LCPUFA-rich food may serve a protective role for AMD patients. Another study illustrated that the extent of CNV can be reduced with the use of APN peptide I utilizing laser-induced neovascularization in the choroidal mice model [15,16,80]. APN is an anti-inflammatory protein and has more than 75% homology with complement protein C1q. The APN1 peptide was derived from the globular region of this protein. APN1 inhibited CNV by more than 75% when injected sub-retinally compared to control peptide [81]. 

We believe that for the purpose of sustained APN1 activity, it can be pegylated for clinical use. Several different doses of APN1 can be utilized in the rodent model to mimic some insight into appropriate human dosing. Once we understand the binding mechanism of APN1 to its receptors, such as AdipoR1, by our experiments or from a literature search, we can design a blocker or antagonist to further proceed and discover whether this blocker or antagonist will also block APN1 binding. It is already known where APN binds to AdipoR1 but not APN1 (which we have designed) [28,55,60]. Currently, we are unsure of how well the APN1 inhibition of wet AMD will work in humans; we expect it will be less invasive as an eye drop or may require less injections with greater interval between treatment visits compared to the drugs that are available on the market today.

## 5. Conclusions

The multifarious beneficial effects of APN/AdipoRs signaling have been exerted in numerous cell types, such as insulin-sensitizing, anti-inflammatory actions, anti-atherosclerotic, anti-carcinogenic, and anti-proliferative effects. Since APN has an ameliorating function on insulin resistance, diabetes, and aging, a reduced APN level is believed to play a vital role in the pathophysiology of retinal diseases and it is associated with the possibility of developing diabetes-associated DR and AMD-associated neovascularization. The research to prove the significant roles of AdipoR1 and AdipoR2 gained momentum due to cloning of these two adiponectin receptors, confirming their requisition for the binding of APN, and subsequently, its glucose-lowering effect. Furthermore, the elimination of APN could lead to the activation of AMPK/SIRT1/PGC-1α and nuclear receptors PPARs through AdipoRs signaling. A screening method for low molecular compounds for AdipoRs agonists using myriad therapeutic approaches along with other potential therapeutic regimes/approaches could be devised utilizing the 3D conformational analysis of AdipoRs. In fact, AdipoRs agonists are optimized based on the 3D analysis of AdipoRs agonist-AdipoRs conformation so as to develop efficacious, safe, and premium class drugs for treating visually disabling diseases. Future research should focus on clarifying AdipoRs and targeting its agonist in order to develop novel anti-aging and anti-diabetic drugs, all while facilitating both the concept of molecular mechanisms’ APN activity and obesity-related and other metabolic disorders. 

Furthermore, AMD, DR, and retinopathy of prematurity are all associated with altered circulating APN levels or APN variant distributions. Experimentally, APN inhibits retinal and CNV defects. As a key glucose and lipid modulator, APN may re-establish metabolic balance. Intervention with ω-3 LCPUFA and the derivatives of fibric acid enhance the levels of APN in the blood. Exercise may exert a positive production of APN systemically, as well as locally, and it plays a protective role in several eye diseases, such as DR, AMD, RP, glaucoma, and light-induced retinal degeneration. Additional planned studies are needed to further investigate and to clarify the role of APN/AdipoRs in DR and CNV, as well as the underlying molecular mechanisms, to better understand both the experimental and clinical impact of this pathway.

## Figures and Tables

**Figure 1 biomolecules-13-00982-f001:**
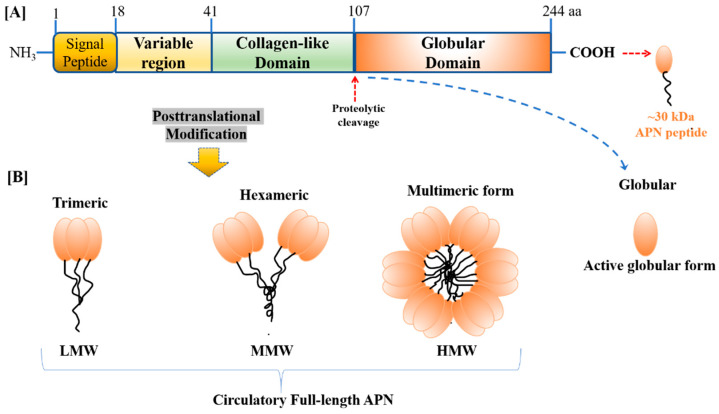
The primary structure of (**A**) human adiponectin (APN) monomer (244 aa and ~30 kDa peptide) consists of four different domains, i.e., NH3-terminal a signal-peptide domain, followed by a short variable region, following a collagen-like fibrous domain, and the long C-terminal globular domain, (**B**) APN monomer undergoes post-translational modification to form four different isoforms present in the bloodstream, i.e., low molecular weight (LMW) trimer, medium molecular weight (MMW) hexamer, high molecular weight (HMW) multimer, and proteolytic cleaved product, i.e., globular APN.

**Figure 2 biomolecules-13-00982-f002:**
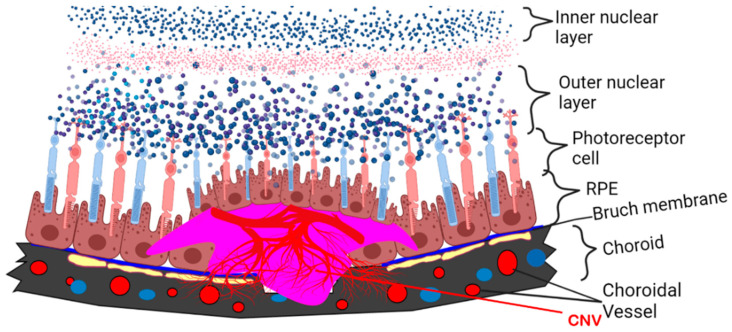
Schematic diagram of laser-induced CNV in mouse model mimics wet type AMD. Red blood vessels indicate new vessels formed under subretinal space.

## Data Availability

Not applicable.

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
