# Peer review of "Emerging Role of Adiponectin/AdipoRs Signaling in Choroidal Neovascularization, Age-Related Macular Degeneration, and Diabetic Retinopathy"

_biomolecules, 2023, doi:10.3390/biom13060982_

Round 1
Reviewer 1 Report (Previous Reviewer 2)
The manuscript has been revised and the quality has been improved. I have the following minor points.
1. The figures in the manuscript might focus on the topic, that is the roles of Adiponectin/AdipoRs Signaling in choroidal neovascularization, age-related macular degeneration, and dia-3 betic retinopathy. The figure 2 and figure 3 show the formation of CNV and the Laser induced CNV in mice, which have been clarified before and are not related the roles of APN in DR and AMD. The author might summarized the involvement of APN in the development of CNV . Figure 3 is a staining of CNV in mice, is the section satined by the author, or it is cited from published paper? The author should clarify it. The figure 4 is not correct and relevant. What is the control means?
2. The language of the revised part is not clear and coherent. It needs proofreading.
Author Response
Please see the attachment.

Reviewer 2 Report (Previous Reviewer 3)
The reviewed version did not adress the concerns of the Reviewer. Dry or wet AMD is not usually causing complete vision loss (as stated in the abstract). As the authors point, the review is an amalgamation of complex data, simple basic data (Figure 2 for instance) with no focus on APN, and self citation (the final paragraphs of the manuscript).
Author Response
Please see the attachment.

Reviewer 3 Report (New Reviewer)
Overall, the manuscript requires much improvement. The structure, format and the use of English language requires upgrading. In addition, the type of the article is not clear, it seems to be something between a review and an original study. Please clarify and re-structure accordingly.
Abstract: not clearly written, needs re-writing, is this a review or an original study? It is not clear from the abstract. Also, avoid the use of the dates (e.g. 1990, 2003) in the abstract, since this is the part of the paper that should consist of summative information. In addition, study design, results and conclusion should be included in the abstract.
Keywords: avoid the use of caps in every word
Introduction: requires better structure, format (avoid the use of caps when unnecessary) and the use of English language requires improvement. At some points, the meaning of the paragraphs is not very clear.
L37: Does not make sense – beneficial pathophysiological??
Figure 1: better quality is required, i.e. the letter in the image should be clear
L241: Repeating about the two different types of AMD
L276: Again, repetition about AMD
Figure 3: is this your work? Or is it another group’s work? If its your work, the procedure should be described in a separate section.
Figure 4: same comment as for figure 3
Round 2
Reviewer 2 Report (Previous Reviewer 3)
The authors introduced their own study in the review, but this part remains the main concern of the reviewer.
Line 205-207 - verb missing (are focusing)
Line 222-224 - rephrase in 2 sentences and correct grammar> ..., or distorted vision. Activation of... , aiding in this condition and improving vision.
PLease insert reference to Figure 3 (own work or bibliography?)
Line 319 - the own study should be properly introduced. A subchapter title is needed
A lot of information is lacking from the study: number of animals, peptide preparation. Animal description. Which mice had Vldlr KO? Why only those developed CNV (laser is a nonspecific method to induce CNV). Was PBS used as control (not clear from the text)?
REsult description has no connection to the methods described. There were intravitreal injections, in results there were discussions only about diet. Line 349 starts with discussion, introducing references. Figure 4 has reference to index 17 from bibliography. It is not clear if the further discussion follows the own study (because authors introduce subretinal injections, while in methods they discuss about intravitreal), or introduces other studies, more likely.
Line 367-375 refer again to author`s study.
Conclusion. Actually, lines 319-388 are self-citation of the reference no. 17 (Bora, P.S.; Kaliappan, S.; Lyzogubov, V.V, et al. Expression of adiponectin in choroidal tissue and inhibition of laser induced 480 choroidal neovascularization by adiponectin. FEBS Lett. 2007, 581, 1977-1982. https://doi:10.1016/j.febslet.2007.04.024.10.1016/j.febslet.2007.04.024), and figure 4 is a readaptation of figure 5 of same text. The presented study seems not a new study, but a résumé, lacking critical information to allow the assesement of validity of the study.
Line 339 - CNV not CVN
Reviewer 3 Report (New Reviewer)
Thank you for considering the suggestions. The manuscript has been improved, however, the style of the review is still mixed - review vs original research. In understand that you use your own research as a supportive data, please clarify where possible, even in the abstract, that you have used some original research to support the literature.
This manuscript is a resubmission of an earlier submission. The following is a list of the peer review reports and author responses from that submission.
Round 1
Reviewer 1 Report
This paper reviews the role of adiponectin and adiponectin receptors in retinal diseases. This is an important topic for review, and the authors are experts on adiponectin and its receptors. However, this paper is inadequate as a timely review of the topic. It is so awkwardly written that it is unlikely that anyone would read it. The review of the literature is not up to date: There were at least a dozen papers on adiponectin and the retina published in 2021 and 2022, and others in 2020 and 2019 that the authors failed to cite. In addition, large sections of the paper were copied from a previous review by the same authors or from other papers, without attribution or quotation marks. Should the authors wish to resubmit a review on this topic, they should start fresh and write new text based on the current literature.
Reviewer 2 Report
The review atricle by Choubey et al summarived the role of Adiponectin/AdipoRs Signaling in the development of choroidal neovascularization (CNV) in the diabetes-associated retinal dystrophy (RD) and wet Age-Related Macular Degeneration (AMD). The review is well written and added something new in the fields. I have the following major points.
1. The title might focus on the CNV in DR and Wet AMD instead of Retinal Complications, as only DR and Wet AMD were dicussed in the review.
2. APN might be selected as therapeutic target for CNV in DR and Wet AMD and some reports in in vitro experiment were cited. I suggest the author briefly disscuss the limitation in these experiments and point out the possiblity of improvement in the treatment. The underlying mechanisms and the compariation with the traditonal anti-VEGF treatment are also important to show the advances in the therapy.
3. As a new treatment in the DR and Wet AMD, I suggest the author briefly discuss the measures to be taken during clinical translational research, for example, how to change the dosage form to improve the long-term effectiveness of APN, and how to design receptor blockers or agonist.
4. Figure 2 might be re-designed , the photoreceptors might be included and the The cell types should be marked on the picture to increase the self-evidence.
Reviewer 3 Report
Line 12-13 - Dry type is also an important cause for loss o central vision, in the andvanced form of geographic atrophy, please correct.
Line 21 - the term distrophy ussually reffers to inherited diseases. Diabetic retinopathy is the term to be used.
Lines 37-40 please reformulate, verb is missing. Line 41 - use . instead of ;
Line 51 - revise [Fu 9]
Line 57-59 - information is repeated from 42-45
Line 157 - please insert reference
Line 197 - current abreviation is VEGF instead of VGEF
Line 211 - please rephrase
Line 233-235 - verb is missing. Use which instead of and (...drusen formation and...)
Line236 - Point is needed befor Dry AMD
Figure 2 - RPE seems to be spliting, which is not possibel (it is single cell layer). Either draw neuroretina (or at least fotoreceptor layer) to show MNV=CNV inside retina (Type 2 MNV) - so redraw top black/dark blue cell layer, or keep RPE intact to show Type 1 MNV.
Line 245 - reference to figure 2 is not needed here
Figure 3 - what is the souce of the figures? What is PBS? You must describe the method used for this picture
Line 262 - addictive? Line 262-263 - please rephrase
Line 266-268 - please rephrase
Figure 4 - does not match the description inparagraph before, where mainly CNV is described and the injection.
This article claims to be a review. However, photos and very incomplete description of a methodology and results (from a personal database, presumably) is provided. It is not clear haw valuable are the new information. The authors have to decide wheter to beter structure a review, or go with a preclinical study.